# Different Dynamics of Bacterial and Fungal Communities in Hive-Stored Bee Bread and Their Possible Roles: A Case Study from Two Commercial Honey Bees in China

**DOI:** 10.3390/microorganisms8020264

**Published:** 2020-02-15

**Authors:** Terd Disayathanoowat, HuanYuan Li, Natapon Supapimon, Nakarin Suwannarach, Saisamorn Lumyong, Panuwan Chantawannakul, Jun Guo

**Affiliations:** 1Faculty of Life Science and Technology, Kunming University of Science and Technology, Kunming 650500, China; lhyuanan@163.com; 2Department of Biology, Faculty of Science, Chiang Mai University, Chiang Mai 50200, Thailand; nattaphon.suphaphimol@gmail.com (N.S.); suwan.462@gmail.com (N.S.); scboi009@gmail.com (S.L.); panuwan@gmail.com (P.C.); 3Research Center in Bioresources for Agriculture, Industry and Medicine, Chiang Mai University, Chiang Mai 50200, Thailand; 4Research Center of Microbial Diversity and Sustainable Utilization, Chiang Mai University, Chiang Mai 50200, Thailand; 5Academy of Science, The Royal Society of Thailand, Bangkok 10300, Thailand

**Keywords:** microbial community, corbicular pollen, Chinese commercial honey bee, next-generation sequencing

## Abstract

This study investigated both bacterial and fungal communities in corbicular pollen and hive-stored bee bread of two commercial honey bees, *Apis mellifera* and *Apis cerana*, in China. Although both honey bees favor different main floral sources, the dynamics of each microbial community is similar. During pH reduction in hive-stored bee bread, results from conventional culturable methods and next-generation sequencing showed a declining bacterial population but a stable fungal population. Different honey bee species and floral sources might not affect the core microbial community structure but could change the number of bacteria. Corbicular pollen was colonized by the Enterobacteriaceae bacterium (*Escherichia-Shiga*, *Panteoa*, *Pseudomonas*) group; however, the number of bacteria significantly decreased in hive-stored bee bread in less than 72 h. In contrast, *Acinetobacter* was highly abundant and could utilize protein sources. In terms of the fungal community, the genus *Cladosporium* remained abundant in both corbicular pollen and hive-stored bee bread. This filamentous fungus might encourage honey bees to reserve pollen by releasing organic acids. Furthermore, several filamentous fungi had the potential to inhibit both commensal/contaminant bacteria and the growth of pathogens. Filamentous fungi, in particular, the genus *Cladosporium*, could support pollen preservation of both honey bee species.

## 1. Introduction

Colony collapse disorder (CCD) has resulted in the decline of the honey bee population worldwide [1]. Pesticides, land use, pathogens, mobile networks are common reasons candidates [2]. It has been proposed that non-pathogenic microbial associations or the microbiome can affect honey bee populations. Dysbiosis (imbalance between an organism and its associated microbial community) of the microbiome can severely affect the health [3], immune system [4], and behavior [5] of honey bees [6] and model animal species [7,8,9]. Numerous studies have investigated the core microbes that exist in the guts of honey bees and their diversity in the same local sites, function, and the effects of dysbiosis. The core community consists of bacteria from nine taxa [10] and some fungi [11] and is established from floral and dietary (hive-stored bee bread) sources.

The floral source is essential for the nutrition of honey bees, in particular, nectar and pollen. Pollen is the most important source of amino acids (proline, arginine, threonine, etc.), carbohydrates (glucose, fructose, lactose, etc.), lipids, vitamins (C, B, E, etc.), enzymes (amylase, pectinase, phosphatase, etc.), and minerals (calcium, magnesium, potassium, etc.) [12]. Foraging bees secrete saliva that is used to moisten and pack the pollen grains onto their hind legs “Corbicular pollen basket”. They carry this to the beehive where forager bees transfer the bee bread into the honeycomb cells to protect it from oxygen[13]. Previously, it was suggested that honey bees favor the consumption of microbially-processed bee bread from action [14,15,16]. Most studies have focused on the roles of gut-associated microorganisms and how microbes convert pollen to bee bread.

Microorganisms that have been associated with honeybees and their food have been identified using two approaches [17,18,19]. Conventional and culture-independent methods have been used to examine the associated bacterial and fungal communities. In bee pollen, three core microbial genera have been identified, *Lactobacillus*, *Pseudomonas*, and *Saccharomyces*, that likely have an important role in the modification of stored pollen. The production of lactic acid from *Lactobacillus* stabilizes bee pollen; however, the roles of *Pseudomonas* and *Saccharomyces* have been well established, but they could be involved in the degradation of pollen walls. Although yeast populations are small in bee pollen, the population size can increase during fermentation [20,21]. Bee bread is known to contain fungal and yeast communities, including *Aspergillus*, *Cladosporium*, *Curvularia, Eupenicillium*, *Fusarium, Gibberella*, *Mucor*, *Penicillium*, *Pestalotiopis*, and *Rhizopus*. The major genera of fungi that have been identified are *Cladosporium*, followed by *Aspergillus* and *Penicillium*. The genera *Mucor* and *Pestalotiopis* have been found in new bee pollen but disappeared after six weeks of storage in the hive [22]. Yeast species that have been identified in pollen and bee bread include *Candida guilliermondii* var. *guilliermondii*, *Cryptococcus albidus* var. *albidus*, *Cr. albidus* var. *diffluens*, *Cr. laurentii* var. *laurentii*, *Cr. laurentii* var. *magnus*, *Kloeckera apiculata*, *Metschnikowia pulcherrima*, and *Rhodotorula pallida* [23]. Land use [24], floral sources [25], and seasonal variation [26] also affect the microbial composition in pollen and hive-storage bee bread. Microbial activity (microbial succession, anaerobic breakdown, and pollen predigestion) improves the nutrition of stored bee bread for honeybees [17,18,23,27,28,29,30,31,32,33]. Furthermore, it has been reported that processed bee bread could have antimicrobial, antioxidant, and anti-radiation properties [34].

In 2014, a new perspective was presented concerning the microbial community associated with pollen and hive-stored bee bread [26]. Using conventional culturable methods, pyro-sequencing, and electron microscopy, it was found that honeybees (foragers and workers) prefer to consume fresh bee bread (no older than 24–72 h) than older hive-stored bee bread [26]. However, this research only focused on the bacterial communities in new pollen and hive-stored bee bread but did not evaluate the role of associated fungal communities.

In Eastern Asia, the Chinese honey bee (*A. cerana*) is used in commercial hives in addition to the Western honey bee (*A. mellifera*). The products of the former are highly valued as medical ingredients and organic food substances [35]. To date, the majority of studies of the microbes associated with Chinese honey bees have focused on core gut-associated bacteria [36]. However, only one study isolated a new lactic acid bacterial species (*Lactobacillus panisapium*) from the bee bread of Chinese honey bees [37].

In this study, we investigated the microbial community in *A. mellifera* and *A. cerana* in China. Both the bacterial and fungal community structures in corbicular pollen and the hive-stored bee bread of each species were evaluated by conventional culture methods and high-throughput sequencing approaches. Our results showed fungal isolation in the new pollen and hive-stored pollen that could suggest a possible interaction between the microorganisms present in the dietary sources of *Apis* bees.

## 2. Materials and Methods

### 2.1. Corbicular and Hive Storage Pollen Collection

This study was performed using samples from *A. mellifera* and *A. cerana* experimental hives (6 hives for each species) located in the same apiary site at the Faculty of Life Science and Technology, Kunming University of Science and Technology (Yunnan, China) during July 2018 (the middle of summer). Colony expansion and queen rearing occurred naturally at the beginning of spring 2018. Corbicular pollen (PC: *A. cerana*; PM: *A. mellifera*) was removed from the legs of 12 bees from each species and hive. In pollen storages’ frames in each hive, stored pollen cell, honey cell, brood cell, and empty cell were labeled with black dots on transparent plastic sheets as undetermined date hive storage pollen 7 (not used in this experiment). Next, after 24 h, in the same hive frame, was investigated the 12 new pollen storage cells and labeled with red dots (< 24 h pollen) and were sealed with clear small capsules with an opened hole to prevent further pollen being added by the bees. At 48 h, all 12 sealed pollen cells were collected as 48 to 72-h hive-stored bee bread (BC in *A. cerana*; BM in *A. mellifera*), which bees prefer to consume [26]. All 12 pollen cells were pulled as one replicated hive and followed this method in 6 hives of both species.

### 2.2. Pollen Identification

Twelve corbicular pollen and hive-storage samples were mixed with a sterile spatula, and their color and pH were recorded. Pollen morphology was compared with floral source pollen that had been collected from 1 km around the apiary site under a compound microscope (Olympus CH30,Tokyo, Japan) to identify the main floral source for each species.

### 2.3. Microbe Number Determination

Each sample (0.25 g) was suspended in 750 μL of 0.85% NaCl buffer with 1% Tween 80 solution and vortexed for 1 min. A 10-fold dilution was made from the suspension, and 200 μL of the diluted sample was spread onto potato dextrose agar (PDA) (incubated for 72 h at 30 °C) or nutrient agar (NA) (incubated for 24 h at 37 °C) to determine the number of fungal and bacterial colonies, respectively. After incubation, the bacterial and fungal colony-forming units (CFU/g) were recorded.

### 2.4. DNA Extraction

Corlicular pollen (PC and PM) and hive-stored bee bread (BC and BM) from each hive were weighed to 0.5 g and added to 1.5 mL microcentrifuge tubes containing a 10% Tween 80 and 0.85% NaCl solution and homogenized. Suspensions were centrifuged at 34 g for 3 min to separate microbes from the pollen, and 300 μL of the separated suspension was transferred to another new centrifuge tube with 700 μL of fresh buffer, performed 8 times and pooled the only suspension from 5th to 8th time (2800 μL) together in 5 mL centrifuge tube that was spun for 10 min at 16,000 g. The supernatant was removed using a pipette, and the pelleted cell was transferred to a sterile blender. Liquid nitrogen was poured onto the pellets, and they blended into a powder. Subsequently, the powder was combined with 750 μL of Bashing Bead^TM^ buffer (ZYMO RESEARCH, Irvine, CA, USA), and DNA was extracted using the quick-DNA Fecal/Soil Microbe Microprep kit according to the manufacturer’s instructions. A final volume of 30 μL of genomic DNA was eluted for downstream analysis.

### 2.5. Amplicon Sequencing Analysis with Illumina Solexa

The concentration and purity of DNA extracts were determined using a NanoDrop 2000 UV-vis spectrophotometer (Thermo Scientific, Wilmington, NC, USA), and DNA quality was evaluated using 1% agarose gel electrophoresis. The V3-V4 hypervariable regions of the bacterial 16S rRNA gene were amplified with primers 338F (5′-ACTCCTACGGGAGGCAGCAG-3′) and 806R (5′-GGACTACHVGGGTWTCTAAT-3′) [38]. The ITS1-ITS2 region of fungal ITS genes were amplified with primers ITS1 (5′-CTTGGTCATTTAGAGGAAGTAA-3′) and ITS2 (5′- TGTGTTCTTCATCGATG-3′) [39] using a thermocycler PCR system (GeneAmp 9700, ABI, Waltham, MA, USA). The PCR reactions were performed using the following cycling parameters: 3-min at 95 °C, 27 cycles of 30 s at 95 °C; 30 s at 55 °C, and 45 s at 72 °C; and a final extension of 72 °C for 10 min. PCR reactions were performed in triplicate in 20 μL mixtures containing 4 μL of 5× FastPfu Buffer, 2 μL of 2.5 mM dNTPs, 0.8 μL of each primer (5 μM), 0.4 μL of FastPfu Polymerase, and 10 ng of template DNA. The amplicons were extracted from a 2% agarose gel, purified using the AxyPrep DNA Gel Extraction Kit (Axygen Biosciences, Union City, CA, USA) and quantified using QuantiFluor™-ST (Promega, Madison, WI, USA) according to the manufacturer’s protocol.

PCR amplicons were sequenced by MajorBio Shanghai Technologies Co., Ltd. (http://www.majorbio.cn, Shanghai, China) using an Illumina MiSeq platform TruSeqTM. The DNA Sample Prep Kit preparation protocol for 2 × 300 bp paired-end reads was used, according to the manufacturers’ guidelines, with 24 samples per lane. Trimmomatic was used to quality-filter raw fastq files and combined by FLASH with 3 criteria: (1) an average quality score < 20 over a 50 bp read; (2) no more than 2 bp mismatch with an overlap longer than 10 bp; and (3) separate barcodes (exactly matching) and primers (allowing 2 nucleotide mismatching). Operational taxonomic units (OTUs) were clustered with a 97% similarity cut-off using UPARSEversion7.1 (http://drive5.com/uparse/) with a novel “greedy” algorithm that performs simultaneous chimera filtering and OTU clustering. The taxonomy of each 16S rRNA gene sequence was analyzed by the RDP Classifier algorithm (http://rdp.cme.msu.edu/) against the Silva (SSU123) 16S rRNA database using a confidence threshold of 70%. The ITS gene sequence was analyzed using the UNITE database (https://unite.ut.ee) [40]. PICRUSt 1.0.0 (http://picrust.github.io/picrust) was used for the functional genomic analysis predictions of the bacteria associated with corbicular pollen and hive-stored bee bread based on the Silva 16S rRNA dataset.

### 2.6. Microbial Isolate Identification

Colonies with different morphologies were randomly selected for growth on fresh media and incubated according to the organism’s growth condition. The vegetative colonies were isolated to obtain single colonies by purification using the streak plate method. Single colonies were transferred to fresh media to determine their biological activity in the next step. The genomic DNA of single isolates of 37 filamentous fungi and 9 bacteria were extracted using ZYMO RESEARCH Fecal/Soil Microbe Microprep (ZYMO RESEARCH, Irvine, CA, USA) following the manufacturer’s protocol. A 25 μL PCR mix was used for fungal genomic DNA that contained 1 μL of fungal genomic DNA, 12.5 μL of 2X TSINGKE master mix buffer (Beijing TsingKe Biotech Co., Ltd., Beijing, China), 6 μL of ddH_2_O, 2 μL of primer ITS1 (5′-TCCGTAGGTGAACCTGCGG-3′), and 2 μL of primer ITS4 (5′-TCCTCCGCTTATTGATATGC-3′) [39]. The following thermal cycling parameters were used: 5 min at 94 °C; 30 cycles of 40 sec at 94 °C, 40 sec at 55 °C, and 40 sec at 72 °C; and a final extension for 10 min at 72 °C. Bacterial genomic DNA was amplified using the PCR mix but with 1 μL of bacterial genomic DNA, 2 μL of primer 27F (5′-AGAGTTTGATCCTGGCTCAG-3′), and 2 μL of primer 1492R (5′-GGTTACCTTGTTACGACTT-3′) [41] (Lane, 1991). The following thermal cycling parameters were used: 5 min at 94 °C; 30 cycles of 40 sec at 94 °C, 90 sec at 58 °C, and 90 sec at 72 °C; and a final extension for 10 min at 72 °C. All amplicons were sequenced by MajorBio Shanghai Technologies Co., Ltd. (http://www.majorbio.cn, Shanghai, China), and the sequences were identified using the National Center for Biotechnology Information (NCBI) (Bethesda, MD, USA) database.

### 2.7. Organic Acid and Hydrolytic Enzyme Screening

Filamentous fungal and bacterial isolates were screened by spotting cells onto 4 different media that indicate enzymatic hydrolysis. Skimmed milk agar (SMA) was used for protease, tributyrin agar for lipase, carboxymethylcellulose agar (CMC) for cellulose, and starch agar (SA) for amylase. Fungi were incubated for 72 h and bacteria for 24 h before recording the clear zones around the spotted colonies on the SMA, tributyrin, and SA plates. The clear zones on the CMC agar plates were recorded after soaking the media in Congo red dye. For organic acid screening, all isolates were spotted onto a mineral agar acid indicator medium and incubated for 3 days. The presence or absence of a yellow zone around the isolates was recorded.

### 2.8. The Antagonistic Inhibition of Filamentous Fungal Isolates with Association Bacteria in the Hive and During Chalkbrood Infection

Chalkbrood (*Ascosphera apis*) infected broods were delivered from the Shaanxi and Guizhou provinces in China. The single spore isolate technique was used to purify both chalkbrood strains to investigate the antagonistic inhibition with bacteria and fungus from pollen. Both strains were identified by amplifying the ITS1-ITS4 gene using the method described for filamentous fungi.

Bacterial isolates were cultured in a 5 mL nutrient broth (NB) for 24 h, and fungi were cultured in potato dextrose broth (PDB) for 3 days. Cultures were spun at 15,000 g in an ultracentrifuge for 15 min, and the pellets were lyophilized until dry. The dried powder was suspended in 500 μL of sterile water for the antagonistic test cross between bacterial and filamentous fungal isolates and the two pathogenic chalkbrood strains.

### 2.9. Statistical Analysis

Univariate data determined normal distribution analysis before determining significant differences of operational taxonomic units (OTUs) between treatments: Average, standard error, T-test, diversity indices, and one-way analysis of variance (ANOVA) (*p <* 0.05). Ordination multivariate analysis included non-metric multidimensional scaling (NMDS) analysis and principal correspondence analysis (PCoA) for determining the distance of the ordination plot between treatments. Bray–Curtis distance and significant differences between treatments were calculated using a non-parametric multivariate analysis of variance (NPMANOVA) with a *P*-value lower than 0.05. All statistical analysis, blog plot, and stack area were performed on paleontological statistics software package for education and data analysis (PAST) statistic programming version 3.23 [42].

To determine the taxon impact on the communities, linear discriminant analysis effect size (LEfse) was examined on the Galaxy application (http://huttenhower.sph.harvard.edu/galaxy), which was measured with a Kruskal–Wallis test (<0.05) and pairwise Wilcoxon test (<0.05). The linear discriminant analysis (LDA) score threshold was 4.0 for bacteria and 3.0 for fungi.

Functional gene prediction was performed using Phylogenetic Investigation of Communities by Reconstruction of Unobserved States (PICRUSt) 1.0.0 (http://picrust.github.io/picrust) to analyze the groups based on clusters of orthologous groups (COGs) [43]. Their impact on the investigated group was analyzed by LEfse with an LDA score greater than 3.0.

### 2.10. Sequence Deposition

The 16s rRNA gene and ITS gene amplicon sequences were deposited to the NCBI database with accession number MK905438-42 and MK910044-78, respectively. Raw data for the 48 sequences obtained by Illumina Solexa Miseq were deposited in bio project number SRR9720852-SRR9720875 for bacteria and SRR9720706-SRR9720729 for fungi.

## 3. Results

### 3.1. Pollen Majority and Colony-Forming Unit Numbers

All pollen samples were compared with pollen obtained from different floral sources around the apiary site using light microscopy. The results of different floral sources accounted for most of the pollen collected from each of the honey bee species. The pollen in both corbicular pollen and hive-stored bee bread from *A. mellifera* was a vanilla-white color (code #F3E5AB) and had more than 50% of pollen similarity with the pollen of *Oxalis* sp. (Oxalidaceae) (Figure 1a). The spheroidal-shaped tricolpate pollen was between 30 and 40 μm. In *A. cerana*, the dominant pollen in corbicular pollen and hive-stored bee bread was bright yellow (#FFAA1D) and appeared to be from *Coreopsis* sp. (Asteraceae) with a majority of around 60% of the total pollen (Figure 1b). This tricopolate pollen had a spheroidal shape and spiked surface and ranged between 40 and 50 μm. There was a significant difference in the pH between the corbicular pollen and hive-stored bee bread from *A. mellifera* and *A. cerana* from pH 6.79 ± 0.03 to pH 5.82 ± 0.02 (*p <* 0.001; *t*-test), and pH 7.01 ± 0.02 to pH 5.90 ± 0.02 (*p <* 0.001; *t*-test), respectively (Appendix A). The CFU/g of filamentous fungi in corbicular pollen and hive-stored bee bread was not significantly different between pollen types or two species (Figure 1a; Appendix A). Although the CFU/g of bacteria was significantly higher for both types of pollen from *A. mellifera*, the CFU/g was significantly lower in hive-stored bee bread than in corbicular pollen in both species (Figure 1a,b and Appendix A); *A. mellifera* (PM 4.23 ± 0.04 to BM 3.85 ± 0.03 logCFU/g; *p <* 0.001 *t*-test), and *A. cerana* (PC 3.76 ± 0.04 to BC 2.99 ± 0.05 logCFU/g; *p <* 0.001 *t*-test).

### 3.2. The Quantitative Data of Illumina Solexa

A total of 1,378,161 high-quality bacterial sequences and 1,421,052 fungal sequences were obtained after filtration. In bacteria, a total of 24 samples from both species and pollen types were analyzed. The read length across 24 samples ranged from 270–530 bp and the read length of fungi samples ranged from 200–459 bp. The total number of reads was aligned into a total of 546 and 2610 operational taxonomic units (OTUs) from bacteria and fungi, respectively, at 97% sequence similarity cut-off.

### 3.3. Microbial Diversity in Corbicular Pollen and Hive-Stored Bee Bread

The average number of bacterial and fungal OTUs in hive-stored bee bread was lower than in corbicular pollen in both honey bee species, but this difference was not significant. However, there was a significant difference in the average number of bacterial OTUs in hive-stored bee bread between the two species (*p =* 0.02; Mann–Whitney) (Appendix A). Alpha diversity indices of OTUs from 24 samples were calculated with Chao, Shannon, and Simpson parameters using the PAST program. All alpha diversity indices were lower in hive-stored bee bread than in corbicular pollen, particularly between bacterial OTUs from both honey bee species (Simpson *p <* 0.05, Shannon *p <* 0.05; Dunn’s posthoc with Bonferroni correction). The fungal OTUs in hive-stored bee bread from both species were also significantly different (Simpson *p <* 0.01) (Appendix A).

### 3.4. Microbial Communities in Corbicular Pollen and Hive-Stored Bee Bread

The most abundant phylum in all combined samples was proteobacteria (92.12%), followed by a small population of phylum Firmicutes (6.66%). The other five phyla, Actinobacteria (0.54%), Bacteroidetes (0.34%), Deinococcus-Thermus (0.073%), Spirochaetae (0.067%), and Fusobacteria (0.042%) accounted for less than 1% of the population (Appendix A).

In the corbicular pollen of both species, *Escherichia-Shiga* was the most abundant bacterial genus accounting for 15% to 50% of the total population in all 12 samples. Bacteria in the genera *Pseudomonas*, *Buttiauxella*, and *Pantoea* were also present in all samples (Appendix A). In the corbicular pollen from *A. cerana*, the genera *Rosenbergeilla* and *Buttiauxella* were more abundant than in the corbicular pollen from *A. mellifera* (*p =* 0.019; unequal variance *t*-test). In contrast, *Paracoccus* and core gut bacteria were dramatically higher in *A. mellifera* (*p =* 0.002; *t*-test) (Figure 2a; Appendix A). Core bacterial species, including *Bifidobacterium*, *Gillamella*, *Snodgrassella*, Apha-1,2, and Firm-4,5, were found in the corbicular pollen from both species.

After 48–72 h of hive-stored bee bread, the proportion of different bacterial communities was different in hive-stored bee bread and corbicular pollen. The proportion of *Escherichia-Shiga* decreased to less than 10% from total bacteria OTUs in all hive-stored bee bread samples (*p <* 0.001; unequal variance *t*-test in both species), and a similar pattern was observed for *Pseudomonas* and *Paracoccus*. In contrast, *Acinetobacter* accounted for more than 25% of the total bacterial communities in hive-stored bee bread but less than 5% in corbicular pollen from both bee species (*p <* 0.004; Mann–Whitney in both species). Core gut bacteria were abundant in both corbicular pollen and hive-stored bee bread, and the population increased in the hive-stored bee bread from *A. mellifera* (*p =* 0.01; *t*-test). Overall, core gut bacteria were more abundant in hive-stored bee bread from *A. mellifera* than *A. cerana* (*p <* 0.001; unequal variance *t*-test), but the populations of *Panteoa* (*p =* 0.002; unequal variance *t*-test) and *Rosenbergiella* (*p =* 0.003; unequal variance *t*-test) were higher in the hive-stored bee bread from *A. cerana* (Figure 2a; Appendix A). The populations of bacteria from the genera *Rosenbergiella*, *Paracoccus*, *Pantoea*, and *Escherichia-Shigella* in hive-stored bee bread were also significantly different between the two species (Figure 2a; Appendix A).

The most dominant fungal phyla based on analysis of the ITS gene was Ascomycota (93.55%), followed by Basidiomycota (5.65%) (Appendix A). Other fungal phyla accounted for less than 1% of the total population, including Zygomycota (0.7%) and Glomeromycota (0.1%). *Cladosporium* was the most abundant genus from phylum Ascomycota and accounted for 15–75% in all 24 samples, with an average proportion of 52.20%. Other genera also in phylum Ascomycota accounted for less than 10% of the total population, including *Botrytis* (6.21%), unclassified family Sclerotiniaceae (4.49%), order Trichosphaeriales (3.19%), and genus *Penicillium* (2.55%), *Aspergillus* (2.00%), and *Alternaria* (1.93%). Other genera accounted for less than 1% of the total population in all samples. Only one Basidiomycete yeast genus, *Rhodosporidium*, accounted for 2.65% of the total microbial community. The data is shown in Appendix A.

The results of the ITS gene did not dramatically change the overview between two bee species and pollen types. Based on these results, *Cladosporium* remained dominant, crossing from corbicular pollen to hive-stored bee bread. Furthermore, the abundance of *Cladosporium* as a proportion of the total population in corbicular pollen was not significantly different between the two honey bee species. The proportion of fungal genera associated with corbicular pollen between the two species was significantly different in the case of genera belonging to the unclassified order Trichosphaeriales (*p =* 0.005; Mann–Whitney) and the genus *Aspergillus* (*p =* 0.013; Mann–Whitney). Furthermore, the proportion of several genera was significantly different between the corbicular pollen and hive-stored bee bread from both honey species, in particular, the genera *Penicillium* and *Aspergillus*. *Rhodosporium* in *A. mellifera* (*p <* 0.05; Mann–Whitney) and the unclassified order Trichosphaeriales in *A. cerana* (*p =* 0.005; Mann–Whitney) were also significantly different between the corbicular pollen and hive-stored bee bread. No significant differences in the composition of the fungal communities were found between the hive-stored bee bread from each species (Figure 2b; Appendix A). Moreover, to confirm significant differences between pollen types and species, multivariate statistics were performed by Beta diversity in the next session.

### 3.5. Beta Diversity and Bacterial Functional Prediction

Multivariate analysis NMDS was used to produce an overview ordination plot from the 16S rRNA gene (Figure 3a) based on Bray–Curtis distance with a stress value of 0.006. Corbicular pollen from *A. mellifera* (PM: sky dots) and *A. cerana* (PC: orange square) was closely related but had few overlaps. NPMANOVA (base on Euclidean distance) was significantly different from the Bonferroni-corrected value (*p* = 0.04). After 48–72 h of hive storage, bee bread from both species (BM: blue plus; BC: red cross) significantly separated from the corbicular pollen (NPMANOVA also revealed a significant difference between PM:BM with a Bonferroni-corrected value *p*= 0.015 and PC:BC with a Bonferroni-corrected value *p* = 0.018).

The ordination plot from PCoA also expressed in the same distance with NMDS (Figure 3). The value from axis 1 of PCoA (65.34%) was examined to identify differences between the groups. An analysis of significant differences was performed as shown for the NPMANOVA (PM: PC *p =* 0.008 unequal variance *t*-test; PM:BM *p <* 0.001 *t*-test; PC: BC *p <* 0.001 unequal variance *t*-test; BM: BC *p <* 0.001 *t*-test).

The results of the NMDS and PCoA ordination plots based on the ITS gene OTUs presented all four groups of investigation with a stress value of 0.08 (Figure 3b). There was no significant difference between community groups from two species by NPMANOVA analysis with Bonferroni-corrected *p*-value. The first axis of PCoA analysis (70.07%) also confirmed that there were no significant differences between all four groups.

To identify the taxa that might affect the microbial community of all four investigated groups, linear discriminant analysis effect size (LEfse) analysis and a one-way ANOVA were used to analyze the eight major, associated bacterial and fungal groups. The results showed that the abundance of several microbial taxa was affected by the type of pollen but not by the species of the honey bee. Bacteria in the genera *Escherichia-Shiga* (LDA = 5.5744; *p <* 0.001), *Paracoccus* (LDA = 5.2834; *p <* 0.001), and *Pseudomonas* (LDA = 4.8598; *p <* 0.001) were affected by corbicular pollen, and *Buttiauxella* (LDA = 4.8711; *p <* 0.001) and *Acinetobacter* (LDA = 5.3892; *p <* 0.001) were affected by hive-stored bee bread (Appendix A). In terms of fungi, *Penicillium* (LDA = 4.6795; *p* < 0.001), *Aspergillus* (LDA = 4.5089; *p <* 0.001), and *Alternaria* (LDA = 4.6169; *p =* 0.048) were affected by corbicular pollen, while only *Cladosporium* (LDA = 5.9206; *p* = 0.32) was affected by hive-stored bee bread (Appendix A).

### 3.6. Functional Prediction from 16S rRNA Gene Data via PICRUSTs Based on COGs

The COG database revealed that the most abundant functional gene groups were signal transduction mechanisms, amino acid transport and metabolism, replication, recombination and repair, and cell wall, membrane, and envelope biogenesis (Appendix A). There was no significant difference across all of the functional genes between corbicular pollen and hive-stored bee bread but several significant differences when comparing each functional gene (Appendix A). A LEfse analysis based on linear discriminant analysis (LDA) was used to investigate which functional genes were affected in all four investigated groups. Several genes were identified that distinguished corbicular pollen from hive-stored bee bread. A group of biogenesis genes was prevalent in the corbicular pollen (PM and PC) group, including those involved in translation, ribosomal structure and biogenesis (LDA = 3.9332), DNA replication, recombination and repair (LDA = 3.8707), posttranslational modification, protein turnover, and associated chaperones (LDA = 3.4607). In hive-stored bee bread, genes that are related to nutritional metabolism were significant, such as genes involved in inorganic ion transport and metabolism (LDA = 3.5519) and amino acid transport and metabolism (LDA = 3.8851) (Appendix A).

### 3.7. Isolates and Their Biological Activities

A total of 37 filamentous fungal isolates consisting of 10 isolates from PM, 5 isolates from PC, 16 isolates from BM, and 6 isolates from BC were identified from the NCBI database. Isolates that had more than 98% nucleotide sequence similarity with isolates in the database were selected (Appendix A). The most common isolates were from the genera *Trichoderma* (seven isolates), *Penicillium* (seven isolates), and *Mucor* (five isolates). This is the first known report of the genus *Trichoderma* in this environment. To confirm their taxonomic identification, a neighbor-joining phylogenetic tree was constructed using *Saccharomyces cerevisiae* as an outgroup (Appendix A).

Interestingly, 32 out of 37 (86%) filamentous fungal isolates produced organic acid after 3 days of incubation (Appendix A). Seventeen isolates had faster growth than a chalkbrood disease on honey bee brood from *A. apis’* two isolates from Shanxi and Guizhou (Appendix A; Appendix A), and four isolates could inhibit the progression of the chalkbrood disease (Appendix A; Appendix A). Few of the isolates produced proteases (five isolates) and lipases (six isolates) (Appendix A), and no isolates produced cellulase or amylase under the conditions used in this investigation.

Nine bacterial isolates were also identified using the NCBI database (Appendix A). A neighbor-joining phylogenetic tree was constructed using *Bacillus subtilis* as an outgroup (Appendix A). Five isolates produced organic acid, five isolates produced proteases, and seven isolates produced lipases (Appendix A). None of the supernatant from the bacterial isolates inhibited the progression of the chalkbrood disease.

After 3 days of culture in potatoes dextrose broth (PDB) supernatants, agar well inhibition was performed using all 37 filamentous fungi isolates and 9 bacterial isolates from pollen (Appendix A). Fourteen filamentous fungal isolates could inhibit the growth of 11 fungal isolates, including those from the genera *Trichoderma*, *Aspergillus*, *Penicillium*, *Cladosporium*, and *Alternaria* (Appendix A). Interestingly, *Trichoderma* and *Penicillium* species could inhibit all 10 isolates of bacteria. The bacterial genus *Acinetobacter* was inhibited the most by the filamentous fungi. In contrast, none of the bacterial isolates could inhibit all 37 filamentous fungal isolates.

### 3.8. Cooperation between Conventional and High Throughput Sequencing Results

Data from high-throughput sequencing provided taxonomic proportion structures and was combined with data from the conventional culturable method that showed their metabolic activity. Combining these methods could reveal possible interactions between microbe associations and their functions in a microenvironment. The four most abundant fungal taxa identified from the analysis of the Illumina sequencing results were matched with fungal isolates obtained via the conventional method (Figure 4a). The genus *Cladosporium* was the most abundant genus in fungal communities and accounted for more than 50% of the total fungal population across all samples. Three isolates from the genus *Cladosporium* that produced organic acid were also found by the conventional method. These could play an antagonistic role in terms of the chalkbrood pathogen and inhibited some bacterial-associated isolates. The genera *Penicillium*, *Aspergillus*, and *Alternaria* were also highly abundant as determined by a culture-independent method, which reveals bioactivity.

In terms of the bacterial community, isolates of three genera matched with the most abundant genera based on Illumina sequencing analysis (Figure 4b). Three *Acinetobacter* isolates were identified using the conventional method that produced organic acid and proteases. Interestingly, the Illumina sequencing results provided evidence that this genus was also abundant in hive-stored bee bread. Species from the genera *Buttiauxella* and *Pantoea* were abundant based on the Illumina sequencing analysis and isolated from these environments; however, neither isolate could produce organic acid but released protease and lipase enzymes.

## 4. Discussion

Generally, honey bees forage between early morning and evening [44]. However, physical factors, such as different geographical sites [45] and seasonal temperatures [46], affect their foraging behavior and the amount of pollen that is stored in hives. To limit the impact of these factors, the two species of commercial honey bees used in this study were kept in the same apiary site, and pollen specimens were sampled at the same time; however, the majority of pollen and the time of collection were different between the two species. *A. mellifera* preferred pollen from *Oxalis* plant species, and *A. cerena* collected more *Coreopsis* pollen. Pollinator species select different floral sources, depending on their morphology and foraging behavior. For example, differences in the foraging distance between the species could have affected the major floral source. *A. mellifera* has a longer wingspan and bigger body than *A. cerana*; therefore, this species could fly for a longer distance and visit more flowers [47]. In addition, bees with different sized bodies might prefer to access flowers of a certain size or shape. Finally, species with different peak foraging times might encounter different floral blooms [48].

Different species of plants have variable chemical compositions [49,50] that can shape the physical properties of pollen, including pH, during microbial-associated activity in floral sources [51]. This study found that the main floral sources of the two honey bee species had corbicular pollen of a different pH. However, after 72 h, the pH of hive-stored bee bread significantly decreased in both species. This finding was consistent with previous studies that have investigated different plant species and floral sources, including samples crossing seven states in the USA [27], almond orchards in California [28], various plant species in Arizona [31], and *Mucor pudica* in Thailand [22]. A decrease in the pH occurs because of the addition of biochemicals from the honey bee’s mandibular and hypopharyngeal glands [52], low pH honey from the foregut honey crop, and the presence of some lactic acid bacteria [28,30,53]. All of these factors contribute to a microenvironment with a low pH.

Abiotic and biotic factors that affect the plant’s microenvironment can also alter the structure of microbial communities on plant surfaces, particularly of those associated with pollen. The microbial populations in this study varied in the corbicular pollen’s two honey bee species, particularly concerning the bacterial population identified using the culturable method. This was supported by a significantly different bacterial OTU derived from the culture-independent method. It has been shown that morphological differences between pollen and the outer layer of different plant taxon affect the microhabitat and the degree of microbial association [54,55]. Plants release phytochemicals, such as volatile compounds [56], disease resistance molecules [57,58], and nutritional compounds in different plants, that might also shape microbial communities [59,60]. After 72 h, the bacterial population, out numbers, and the alpha diversity indices in hive-stored bee bread decreased significantly for both species; however, the fungal population remained stable. Anderson et al. [26] reported at least a 10-fold reduction in the bacterial population of hive pollen after 96 h of storage and more than 60% reduction in 24 h, suggesting that these conditions do not promote bacterial growth. In terms of the fungal population, a study that evaluated the fungal population in hive-stored bee bread over 6 weeks in Thailand [22] reported a gradual decrease in the population. In particular, a less than 1-fold decrease was reported in the first week. Low pH environments have a high concentration of hydrogen ions [61], which limits bacterial growth but supports fungal growth, owing to differences in carbon utilization and cellular respiration. In acidic conditions, fungi predominantly use glucose-induced respiration [62]. Furthermore, it has been reported that some filamentous fungi, such as *Penicillium* sp. [63] and *Aspergillus niger*, are capable of maintaining ion homeostasis in the cytoplasm under low pH conditions [64]; however, the opposite has been reported for bacteria results [65]. This evidence is in agreement with the findings presented here and suggests that the hive-stored bee bread environment might support fungal growth while reducing the total bacterial population.

At this point, we have suggested that different species and floral sources contain microbial communities with different population structures. The communities of bacteria and fungus identified from high throughput sequencing might provide more detail on the dynamics of the microbial communities that occur from corbicular pollen to hive-stored bee bread.

A large proportion of bacteria present in the corbicular pollen from both honey bee species belonged to the Gamma-proteobacteria phylum, including the genera *Escherichia-Shiga*, *Pseudomonas*, and *Pantoea.* These three genera are commonly found in plant material, such as fruit surfaces [66], floral nectar [67,68], petals [69], and leaf surfaces [70,71,72]. This population of bacteria has likely optimized its physiological mechanisms for the rich, nutritional environment of flowers, which provides an excellent habitat for many microorganisms. However, plants employ defense mechanisms that alter the microenvironment, such as high osmotic pressure, low oxygen levels, and the production of antimicrobial compounds. *Pseudomonas* and *Pantoea*, which were isolated from floral nectar in Spain [68], showed growth under microaerobiosis and sucrose tolerance. Furthermore, the catalase activity of these bacteria was maintained on exposure to certain toxic substrates from plants [73]. Interestingly, this study found a relatively high proportion of the human pathogen *Escherichia-Shiga* in the corbicular pollen. Generally, this genus is common in the environment surrounding beehives and can contaminate plant material via water or wind [74,75]. However, not all *Escherichia* are coliform pathogens from the mammalian gut. In 2013, Meric et al. [76] identified distinct *Escherichia* strains that were plant- or gut-associated by MLST-based phylogenetics and physiological methods [77]. The plant-associated strains were better adapted to survive on plant material, owing to increased biofilm and extracellular matrix production [74,78,79]. Furthermore, these bacteria were able to utilize sucrose and raffinose, which are generally found in plant material [80]. According to Anderson et al. [26], few core gut bacteria are found in corbicular pollen. It was suggested that core gut bacteria are not obtained from the pollen but instead from the social interactions between nestmates [19]. The bacterial communities in the corbicular pollen of the two honey bee species were significantly different but not extremely distinct in this study (NPMANOVA, *p* = 0.04). The two honey bee species were in the same location and season but had different primary floral sources; therefore, we suggested that the corbicular pollen shared common bacterial taxon but had different population structures depending on the species. Furthermore, the proteobacteria found in this study were also reported to digest plant material [81,82,83], suggesting that these bacterial groups might be needed to utilize the rich nutrients of corbicular pollen.

The hive-stored bee bread from both honey bee species had different bacterial communities than in corbicular pollen. Except for a lower pH, the high concentration of sugar that is added by the honey bee likely creates an extreme microenvironment that could reduce the bacterial population. Interestingly, *Acinetobacter*, aerobic genera that are found in many environments, was scarce in the corbicular pollen, but the population increased in the hive-stored bee bread. This microbial-association has been reported commonly in corbicular pollen [67,84] and hive-stored bee bread [26,85]. Two species of *Acinetobacter* have been found as novel species in the nectar of a wild plant in the Mediterranean: *Acinetobacter nactaris* and *Acinetobacter boissieri* [86]. Furthermore, *Acinetobacter apis* has been discovered in the intestinal tract of *A*. *mellifera* [87]. Many studies have shown that *Acinetobacter* species thrive in a sugar-rich habitat and have even formed a symbiosis with the Pae aphid, a phloem sap-eating insect [88]. Moreover, *Acinetobacter* isolates from floral nectar required a high concentration of sucrose for growth and used this sugar to produce a biofilm-like material [67]. Several studies have reported the isolation of this genus in extremely low water [89] and high acid conditions [90,91]. Three strains of this genus were able to release organic acid, which suggests that *Acinetobacter* has adapted to survive in the environment that is common to hive-stored bee bread. In addition, several other core gut bacteria were found in the hive-stored bee bread of *A*. *mellifera* because bacteria, particularly Firm-4,5, are transmitted from the bees into the pollen [10]. *Lactobacillus kunkeei* is another core gut bacteria that has been frequently detected in the foregut and hindgut of *A*. *mellifera* [19] and also hive-stored bee bread [25]. These bacterial species could potentially inhibit the growth of other microbes by producing bioactive compounds [25]. Previous studies that have compared the core gut bacteria between *A. mellifera* and *A. cerana* have also reported that although the same taxa form the microbiota, the communities have different structures [19,36,92,93]. The decrease in *Escherichia-Shiga, Pseudomonas*, and *Pantoea* and the increase in *Acinetobacter* number in hive-stored bee bread affect the dramatic distinction between the bacterial communities in both species. This suggests that honey bees create an extreme environment to protect hive-stored bee bread from contamination by bacteria. However, as highlighted by the results of the functional prediction, some bacterial species can adapt their physiology to survive in these conditions and utilize the available food sources.

The functional analysis of the 16S rRNA gene based on the COG database revealed high scores for genes involved in the biogenesis and bacterial growth. This suggested that corbicular pollen provides a food habitat for the growth and reproduction of bacteria, particularly the phylum proteobacteria. This gene group also decreased following a reduction in commensal/contaminant proteobacteria. Nutritional metabolism genes were dominant in hive-stored bee bread, as shown by higher *Acinetobacter* OTUs, especially the amino acid metabolism groups. Many studies have reported that *Acinetobacter* has the potential to produce proteases [94,95]. Consistent with this suggestion, the culture-dependent results of this study showed that three strains of *Acinetobacter* isolated from hive-stored bee bread released protease enzymes. In addition to the protease enzyme, a biofilm material composed of carbohydrates and polypeptides is also commonly formed by *Acinetobacter* [67,96,97,98]. Generally, hive-stored bee bread is a good source of protein and minerals. These findings suggested that *Acinetobacter* has evolved in this environment to utilize the rich amino acid source available in hive-stored bee bread.

Core communities and the analysis of dynamic ITS genes showed a different perspective of the 16S rRNA gene results. This study identified the dominance of the fungal communities associated with the honey bee’s food source, which were distinct from the gut-associated fungal communities that include genera, such as *Saccharomyces*, *Zygosaccharomyces*, and *Candida* [11]. This study found that the genus *Cladosporium* was abundant in all samples from both honey bee species. Globally, *Cladosporium* is mostly found in the air [99] and associated with plant material, particularly plant pathogens [100]. It is not generally found in flowers [101] and has only been identified in the floral pollen of *Prunus dulcis* in the USA [18]. However, this genus has been investigated as epizootic microbes associated with numerous arthropods, particularly insects, and used as biocontrol pesticides [102,103,104,105]. Unlike bacteria and yeast, which are frequently reported as a two-way transmission between floral sources and pollinators [101,106], mold is different. Martinson et al. 2012 [107], reported that the fungal communities of fig (*Ficus* spp.) flowers changed after pollination by wasps, which suggested fungal transfer from the pollinators to the flower. This study also proposed that thousands of fungi are not necessarily from floral sources but could come from other environments and cohabit with the pollinator, including the genus *Cladosporium*. Currently, it is not clear which microbes are associated with the surface of honey bees. After floral pollen has been collected, *Cladosporium* may be transmitted from plants or bees to corbicular pollen and persist in hive-stored bee bread. According to previous reports, this genus is commonly found in corbicular pollen and hive-stored bee bread worldwide from both culturable and molecular approaches [18,22,26]. *Botrytis* and *Penicillium* are different from *Cladosporium*. Both genera are found in plant pollen and might be transmitted to pollinators [108] in addition to being frequently found in hive-stored bee bread [22].

In contrast to the dynamics of the bacterial community, the fungal community showed a stable community structure from corbicular pollen to hive-stored bee bread (Figure 2 b and Figure 3b). As previously mentioned, fungi favor low pH environments. Several reports have shown that the genus *Cladosporium* is acidophilic and an acid-tolerant fungus [109,110] (Gross and Robbins, 2000; Bridziuviene and Levinskaite, 2007). Moreover, high LDA scores in hive-stored bee bread suggested that *Cladosporium* could evolve to affect this environment. The results from the culture-dependent method also showed that two isolates of the genus *Cladosporium* from hive-stored bee bread (BM9 and BC2) and other filamentous isolates released organic acid (Figure 4a and Appendix A; Appendix A). Moreover, this genus is known to be osmotolerant in various conditions [111,112,113] and has a high sugar tolerance [114,115]. Both physiological adaptations, including acid resistance/acid producer and osmotolerance, suggest that *Cladosporium* might be selected by honey bees to promote a low pH environment and reduce contaminant/commensal Enterobacteria communities.

This study also investigated the antagonistic relationship between filamentous fungi and honey bee brood disease, *Ascospheara apis*. Several isolates had faster growth and inhibited the disease, including one *Cladosporium* isolate (BM9). Faster growth, or monopolizing, is the interaction between organisms in ecosystems. Organisms that can utilize more nutrients in their habitat reduce the growth rates of other organisms. In this microenvironment, mold spores have the potential to germinate [116], including chalkbrood spores. In this study, six *Trichoderma* isolates were monopolized with the chalkbrood disease. This genus is a well-know biocontrol method for bacterial and fungal diseases in agriculture [117,118], owing to their excellent ability to colonize different habitats [119,120]. This genus also inhibited the growth of dominant bacterial isolates from pollen and hive-stored bee bread.

Gilliam et al. (1989) [18] showed that *Aspergillus*, *Mucor*, and *Rhizopus* isolated from bee bread had the potential to inhibit the growth of the chalkbrood disease. In this study, we found that other genera from pollen and hive-stored bee bread also inhibited growth, including *Alternaria, Cladosporium*, *Penicillium*, and *Taralomyce*. Furthermore, *Cladosporium* and *Penicillium* also inhibited bacterial isolates in this environment. We suggested that *Trichoderma* and the other filamentous fungi might play a role in making honey bee hives less favorable for commensal/contaminant microbes through an antagonistic relationship and the release of organic acid.

In conclusion, two species of honey bee that had different floral sources had similar core communities of microbes but different community structures. We suggested that hive geographical sites might impact the microbial community through pollen more than species and floral source factors [85]. The results of this study could support the pollen preservation hypothesis of Anderson and colleagues (2014) [26]. However, we suggested that filamentous fungi might help honey bees to manage their food source. From contact with floral sources, the honey bees obtained microbes from the flowers (Enterobacteriaceae group and *Acinetobacter*) and other parts of the plant and environment (*Cladosporium*, *Trichoderma*). The Enterobacteriaceae favored the utilization of corbicular pollen; however, honey bees controlled this by adding honey, which creates an environment that *Acinetobacter* have adapted to survive in. Moreover, *Cladosporium* and other filamentous fungi also survived in hive-stored bee bread, encouraging honey bees to preserve pollen by releasing organic acid, which is effective at reducing contaminants/commensal microbes (Figure 5). This relationship might occur within 72 h of hive-stored bee bread, which is the honey bee’s preferred dietary source. After 72 h, we suggested that the abundant microbes, especially *Cladosporium* and *Acinetobacter*, would utilize unfinished hive-stored bee bread as their food source because of their ability to utilize amino acids.

This study has provided an overview of microbial interactions in two honey bee food sources predicted through genomic and phenotypic analysis. More information could be derived from platforms, such as transcriptomic or proteomic approaches, in future work, to increase our understanding of microbes’ roles in honey bee food sources. We anticipate that our findings might contribute to efforts to prevent the further decline of honey bee populations, which is a considerable problem for humans worldwide.

## Figures and Tables

**Figure 1 microorganisms-08-00264-f001:**
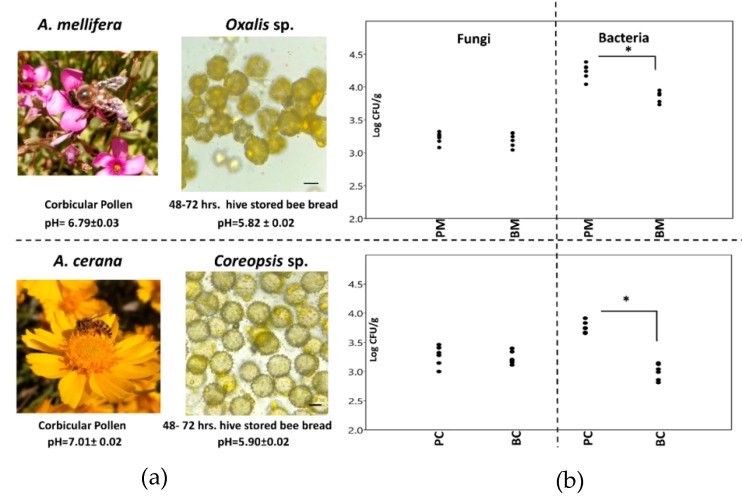
(**a**) The major pollen sources found in the corbicular pollen and hive-stored bee bread of two honey bee species. (**b**) The CFU/g of fungi and bacteria number in corbicular pollen and hive-stored bee bread with an asterisk, indicating a significant difference between the groups (*p* < 0.05). CFU: colony-forming unit.

**Figure 2 microorganisms-08-00264-f002:**
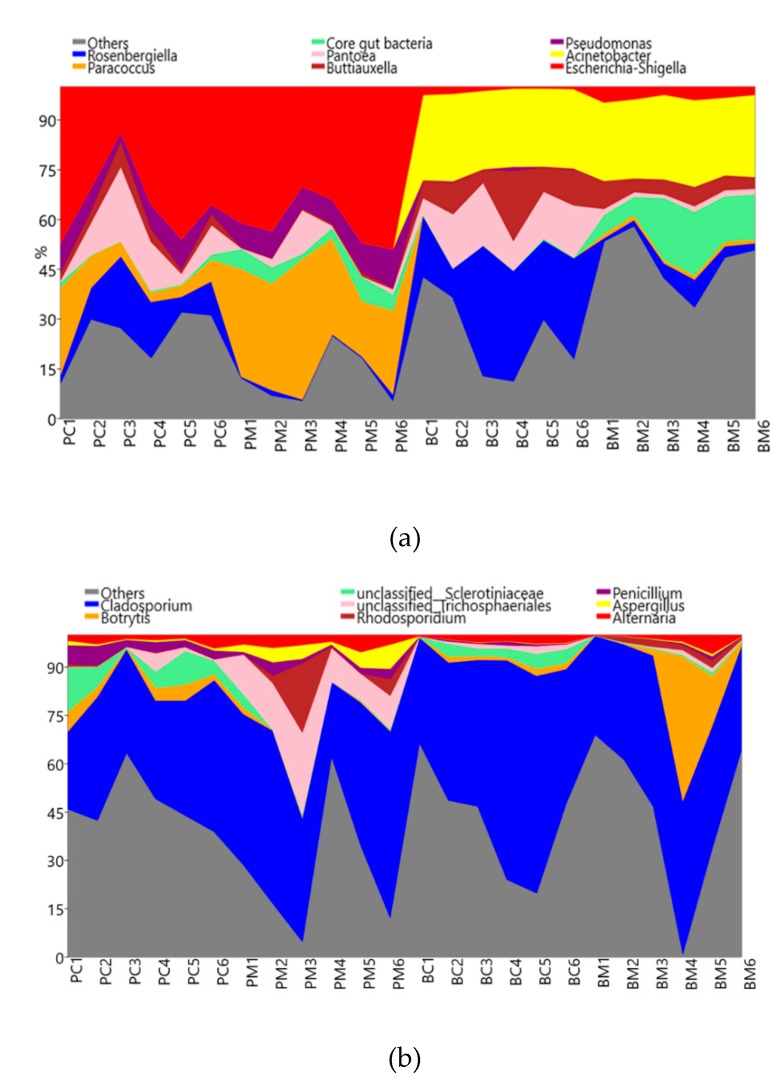
Stack area graph, showing the dynamics in the structure of bacterial (**a**) and fungal (**b**) communities in corbicular pollen and hive-stored bee bread in two honey bee species.

**Figure 3 microorganisms-08-00264-f003:**
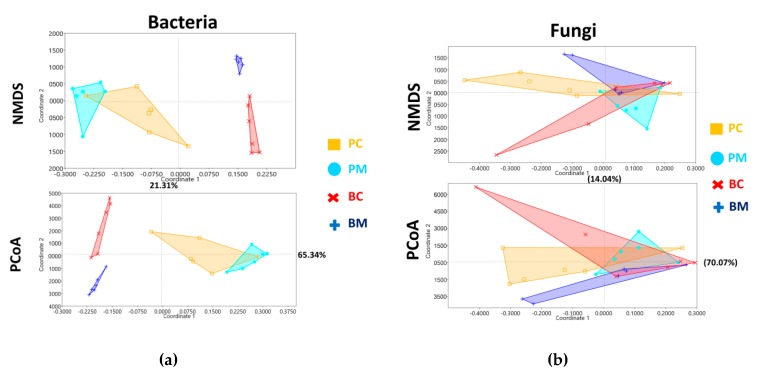
Four ordination plots from non-metric multidimensional scaling (NMDS) and principal correspondence analysis (PCoA) analysis, showing the distance between the investigated groups of bacterial (**a**) and fungal (**b**) community structures.

**Figure 4 microorganisms-08-00264-f004:**
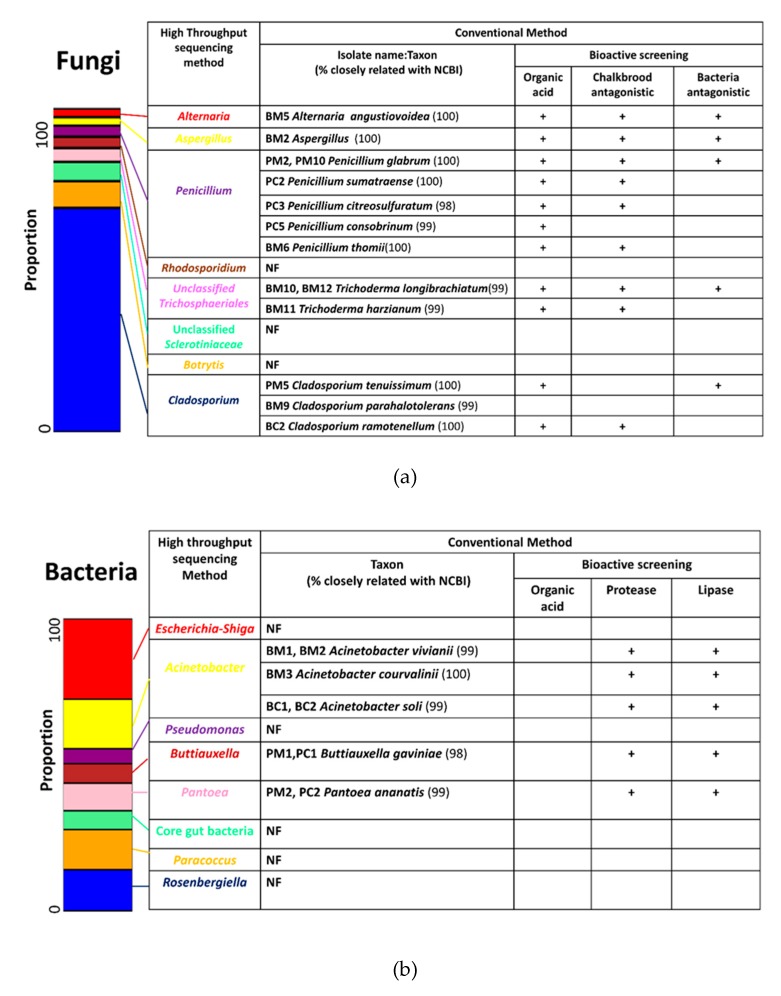
Combined results of high-throughput sequencing with conventional methods for bacterial (**a**) and fungal communities (**b**).

**Figure 5 microorganisms-08-00264-f005:**
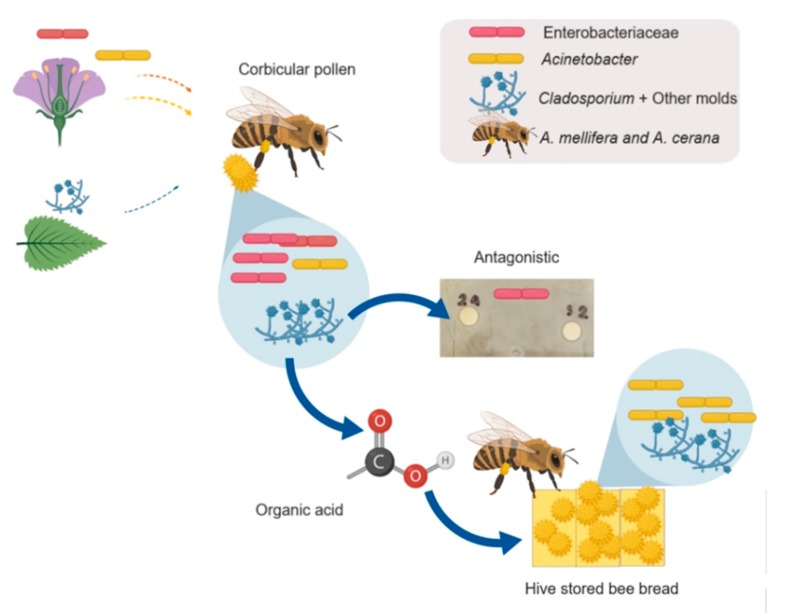
Possible routes and roles of dominant microbes associated with hive-stored bee bread in 2 honey bees.

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
