# Peer review of "Different Dynamics of Bacterial and Fungal Communities in Hive-Stored Bee Bread and Their Possible Roles: A Case Study from Two Commercial Honey Bees in China"

_microorganisms, 2020, doi:10.3390/microorganisms8020264_

Round 1

Reviewer 1 Report

Cannot be reviewed. Rewrite with reference to the english language and style. appears that a synonym reference was used in the writing and really confuses context in many situations.

Almost every sentence requires English revision.

Author Response

Dear Reviewer 1

Regards

Authors

Reviewer 2 Report

Dear Editor and Authors,

Disayatanoowat et al. present the comparative results of bacterial and fungal community dynamics in pollen (corbicular and in the hive) from two bee species in China. Several analyses have been performed (species identification diversity variation over time, the antimicrobial action of the isolates) and all results are essential, mainly due to the decline of these important pollinators across the planet. However, the article presents grammatical and structural problems that prevent a more detailed analysis of the results. Several passages are complicated to understand, and the article deserves an extensive review by a native English speaker. The objectives are not explicit; as a result, it is not possible to associate the methodology with the objectives of the work. I suggest the authors rewrite the article taking into account the purposes of their work. While recognizing the importance of the work, I cannot recommend it for publication in the present form.

Below, it is listed some minor problems, which show the need for a grammatical and stylistic revision so that the work can be appropriately evaluated.

Minor concerns

Abstract

L28 – “but are impact on” or “but have an impact on”?

L29 – “bacteria” instead “bacterium”;

L29 - I would use "h" and "min". I think unit symbols are unaltered in the plural. Please, very the rules of the Microorganisms;

Introduction

L41-42 – this phrase is meaningless;

L48 – “focused on their diverse” or “focused on their diversity”?

L52 – “source of proteins” but are cited amino acids;

L59-60- this phrase is worthless;

L62 – via two;

L83 – results revealed;

Material and Methods

L122 – As a matter of style, it is preferable to begin the phrase with “Two hundred microliters”;

Results

L220 – As a matter of style, it is not recommended to use an abbreviated name for the genus at the beginning of the phrase;

L262-263 - diversity analysis should be detailed in “Materials and Methods” section; Several statistical analyses are presented, but they were not detailed in the MM section. This is a problem of the manuscript due to the confusing (or absent) presentation of the objectives.

L275 – bacteria in genera;

L276 – genera.

Author Response

Dear Reviewer 2

Regards,

Authors

Reviewer 3 Report

135-136 line - 30 μl of final 135 genomic DNA was eluted for analysis. No information about DNA concentration.

177 line: is 72oC; Should 72°C

 Supplementary material Tables

I propose to present tables 1 - 4 in the form of figures.

Tables 1-3: Complete the empty Fields (Sample), below in column 1, 2, etc

Table 5 is actually two tables. Please indicate 5a and 5b respectively.

Table 6: COG -  clusters of orthologous groups (please add)

Table 7: Delete the last two blank lines.

Insert information - shaded field means - presence / activity

Applies to tables 7, 8 and 9.

The headings in the tables are once uppercase and once lowercase. Unify the record.

Table 8: Complete / or remove columns: acid , Antagonist with Chalkbrood - Faster growing / inhibition growing

Author Response

Dear Reviewer 3

Regards,

Authors

Round 2

Reviewer 2 Report

Dear Editor and authors,

The authors have presented an improved version of the manuscript in a concise language. The results will be important for improving the quality control process of honey production.

Sincerely yours,

Author Response

Dear Reviewer 2, 

  Massive Thanks for your kind suggestion.

We also hope our contribution will benefit on apicultural field. 

Regards,

Authors